# Application of Fuzzy TOPSIS to Flood Hazard Mapping for Levee Failure

**Tae Hyung Kim [1], Byunghyun Kim [2],\* and Kun-Yeun Han [3]**

[1] Forecast and Control Division of Nakdong River Flood Control Office, Ministry of Environment, Pusan 49300, Korea; kimth3515@korea.kr

[2] National Civil Defense and Disaster Management Training Institute, Ministry of the Interior and Safety, Cheonan 31068, Korea

[3] Department of Civil Engineering, Kyungpook National University, Daegu 41566, Korea; kshanj@knu.ac.kr

\* Correspondence: bhkimc@korea.kr; Tel.: +82-41-560-0068

**Abstract:** This paper proposes a new approach to consider the uncertainties for constructing flood hazard maps for levee failure. The flood depth, velocity, and arrival time were estimated by the 2-Dimensional model and were considered as flood indices for flood hazard mapping. Each flood index predicted from the 2-D flood analysis based on several scenarios was fuzzified to reflect the uncertainties of the indices. The fuzzified flood indices were integrated using the Fuzzy TOPSIS (Technique for Order of Preference by Similarity to Ideal Solution), resulting in a single graded flood hazard map. This methodology was applied to the Gam river in South Korea and confirmed that the Fuzzy MCDM (Multiple Criteria Decision Making) technique can be used to produce flood hazard maps. The flood hazard map produced in this study compared with the current flood hazard map of MOLIT (Ministry of Land, Infrastructure and Transports). This study found that the proposed methodology was more advantageous than the current methods with regard to the accuracy and grading of the flood areas, as well as in regard to an integrated single map. This report is expected to be expand upon other floods, including dam failure and urban flooding.

**Keywords:** flood hazard map; 1-D and 2-D analyses; Fuzzy TOPSIS method; levee failure

## 1. Introduction

Hazard maps aim to provide residents with information on the range of possible damage and disaster prevention activities [1]. Flood hazard maps are among the representative nonstructural measures for reducing flood potential. Such maps are produced and utilized in different forms (i.e., flood trace maps, emergency action plan maps, and flood information maps) and for diverse purposes. Flood hazard mapping studies have been conducted in various countries as national research projects. In the United States, flood hazard mapping is an important part of the National Flood Insurance Program (NFIP). Further, Flood Insurance Rate Maps (FIRMs) are maintained and updated by FEMA's (Federal Emergency Management Agency) flood hazard mapping program, which are called Risk MAP (Mapping, Assessment, and Planning) [2]. These maps are being released through their public website. In Europe, many countries and organizations are cooperating to establish a joint research system for the construction of flood hazard maps for the rivers that flow through several countries. Typical examples include the European Exchange Circle on Flood Mapping (EXCIMAP) [3] and the Danube Flood Risk Project in Eastern Europe [4]. In South Korea, flood hazard maps are produced by the central and local governments for extreme floods due to dam or levee failure [5]. As such, flood hazard maps are being produced in various forms in many countries depending on the need.

Although flood hazard maps can be produced by multiple methods, the basic process is to use the results of inundation analysis from 1-D or 2-D hydraulic models. A 2-D inundation model is more useful than a 1-D model for creating sophisticated flood hazard maps that are used to forecast flood damage in protected lowlands due to levee failure or overtopping. This is because 2-D models can produce spatially distributed information including flood depth and velocity across the flooded area. Various 2-D inundation models, including CCHE-2D (Center for Computational Hydroscience and Engineering) [6], FLO-2D [7], SOBEK [8], MIKE Flood [9], JFLOW [10], and LISFLOOD [11] have been used to construct flood hazard maps.

Flood damage can vary in size and range depending on the amount of rainfall, the failure shape of hydraulic structures, and the duration of the failure. Therefore, when carrying out a 2-D inundation analysis for flood hazard mapping, various scenarios should be considered to reflect these uncertainties. There have been many studies on risk analysis, reliability analysis, and probabilistic methods (e.g., Monte-Carlo simulation) that reflect these uncertainties in flood hazard mapping [12–15]. This study suggests a methodology for flood hazard mapping using Fuzzy MCDM (Multiple Criteria Decision Making) techniques among various methods.

So far, the Fuzzy MCDM technique has been used more often in flood risk analysis and flood vulnerability analysis than flood hazard mapping. Qin et al. [16] and Jun et al. [17] analyzed flood vulnerability using the MCDM, while Fernandez and Lutz [18] applied the same technique to the analysis of flood risk. In recent years, the studies using specific methodologies such as TOPSIS or fuzzy logic has emerged to create flood hazard map. Arabameri et al. [19] compared several statistical methods and MCDM techniques for flooding hazard mapping. Kanani-Sadat et al. [20] presented a framework for the creation of flood prone areas' maps by integrating GIS (Geographic Information System), fuzzy logic, and MCDM. In addition, Cheng et al. [21], Dai et al. [22], Afshar et al. [23], and Wang et al. [24] used the TOPSIS technique in their fields of water resources.

Despite various studies considering uncertainties in flood hazard mapping, the flood hazard maps currently being created for public purposes in South Korea do not consider the uncertainties of multiple flood scenarios. Even if various flood hazard maps are produced for multiple scenarios, these maps are not integrated because they do not consider the scenarios occurring simultaneously. In Korea, flood hazard maps are created by overlaying the flood depth and extent derived from flood analysis using numerical models. Several flood hazard maps are created in the same area depending on the flood frequency and the uncertainties of flood indices are not considered in flood hazard mapping, as mentioned above. This study aims to improve the methodologies that are currently used to create flood hazard maps. Thus, this study proposes a new approach for integrating the TOPSIS method (which is an MCDM) and fuzzy logic to consider uncertainties in a more rational way via the scenario-based flood hazard mapping using 2-D inundation model.

This paper is composed of the following sections: Section 2 describes our study's flow chart, the differences of other existing studies, the theory of application models, the newly proposed fuzzification, and the Fuzzy TOPSIS techniques for flood hazard mapping. Section 3 shows several scenario-based 2-D flood analysis results for the study area. Section 4 examines the applicability of the methods proposed in this study by comparing flood-hazard map created in this study with those currently in use for public purposes. Section 5, thus, offers conclusions for the study.

## 2. Methodology

### 2.1. Method of This Study

As shown in Figure 1, 2-D inundation analysis was carried out to predict the flooding indices, including depth (H), velocity (V), and flood wave arrival time (T), due to the levee failure or overtopping. The overflow or failure discharge of levee was estimated using FLDWAV (Flood Wave) [25]. The boundary conditions and lateral inflows of FLDWAV were calculated by HEC-HMS (Hydrologic Modeling System) [26]. A 2-D inundation model was applied for flooding analysis due to

levee failure or overflow. Flood depth (H), velocity (V), and arrival time (T) were estimated using the 2-D model and were considered as flood indices for flood hazard mapping. Each flood index predicted from the 2-D flood analysis based on several scenarios was fuzzified to reflect the uncertainties of indices. The fuzzified flood indices are integrated by the Fuzzy TOPSIS technique, resulting in a single graded flood hazard map.

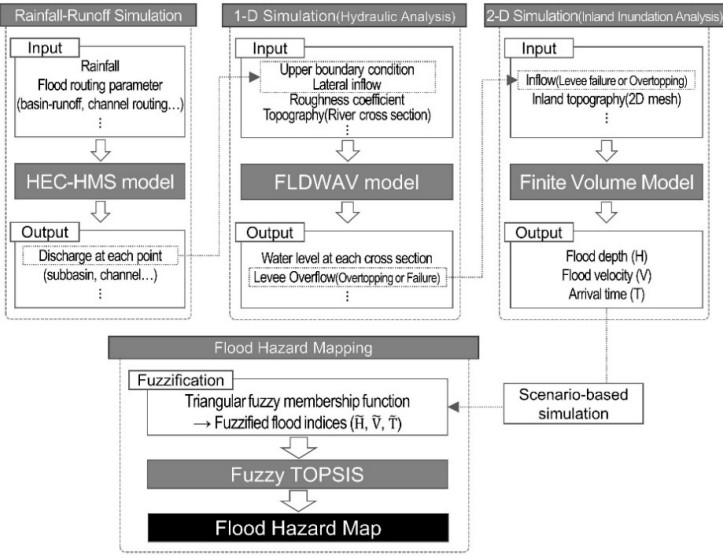

**Figure 1.** Flow chart of this study.

Figure 2 shows the methodological differences for flood hazard mapping between the traditional and the proposed. The traditional approach involves performing a 2-D inundation analysis of the scenarios depicting different frequency or levee failure factors. Then, the inundation forecasting maps are simply overlain according to the inundation depth and velocity only. On the other hand, this study employed fuzzification by considering various input factors for each event scenario, such that all the uncertainties were included in the calculation process. Moreover, the scenarios were not simply overlain according to a single input index. Rather, the flood hazard map was created using Fuzzy TOPSIS to include the uncertainties in other input indices, such as the velocity and flood wave arrival time, in addition to the inundation depth. To verify the applicability of the proposed methodology, a flood hazard map was created for the Gam river and the Jikjisa river (the first and second tributaries of the Nakdong river, respectively) in Gimcheon, South Korea.

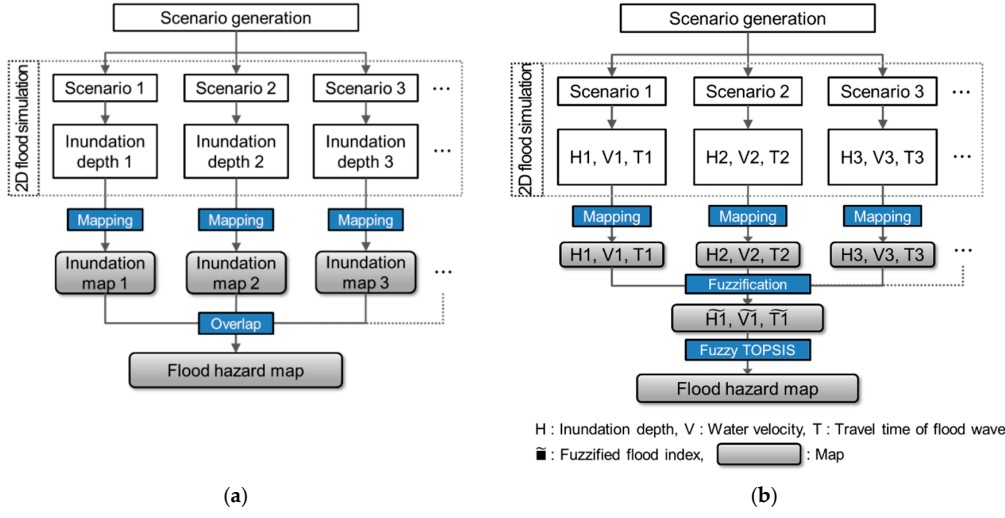

**Figure 2.** Methodologies for flood hazard mapping: (**a**) Traditional and (**b**) Proposed

### 2.2. Rainfall-Runoff Model and 1D Hydraulic Model

The HEC-HMS (United States Army Corps of Engineers, Davis, CA, USA) simulates the rainfall-runoff process, which has evolved from popular US Army Corps of Engineers' HEC-1 model. Its capabilities involve precipitation, rainfall losses, runoff transformation, and channel/reservoir routing [26]. In this study, observed precipitation data, SCS curve number, and Clark unit hydrograph method are applied to estimate the lateral inflows for each tributary.

FLDWAV is a generalized dynamic wave model for one-dimensional unsteady flows in a single or branched waterway [25]. It is based on an implicit (four-point, nonlinear) finite-difference solution of the Saint-Venant equations [25,27]. The continuity and momentum equation can be expressed as in Equations (1) and (2), respectively:

$$\frac{\partial Q}{\partial x} + \frac{\partial (A + A_0)}{\partial t} - q = 0 \tag{1}$$

$$\frac{\partial Q}{\partial t} + \frac{\partial (\beta Q^2 / A)}{\partial x} + gA\left(\frac{\partial h}{\partial x} + S_f + S_e\right) - \beta q v_x + W_f B = 0 \tag{2}$$

where, $x$ is a longitudinal distance along the channel or river, $t$ is time, $A$ is cross-sectional area of flow, $A_0$ is cross-sectional area of off-channel dead storage (contributes to continuity, but not momentum), $q$ is lateral inflow per unit length along the channel, $h$ is water surface elevation, $v_x$ is the velocity of lateral flow in the direction of channel flow, $S_f$ is friction slope, $S_e$ is eddy loss slope, $B$ is the width of the channel at the water surface, $W_f$ is wind shear force, $\beta$ is momentum correction factor, and $g$ is the acceleration due to gravity.

### 2.3. 2-D Flood Model

In this paper, a 2-D high-accuracy finite volume model, developed by this study's co-authors [28,29], was applied to flood hazard mapping based on levee failure and overtopping. To accurately process the irregular topography, an algorithm for nonstructural and hybrid grids was applied [28]. The conservative form of the 2-D shallow water equation can be expressed as in Equations (3) and (4):

$$\frac{\partial \mathbf{U}}{\partial t} + \frac{\partial \mathbf{F}(\mathbf{U})}{\partial x} + \frac{\partial \mathbf{G}(\mathbf{U})}{\partial y} = \mathbf{S}(\mathbf{U}), \tag{3}$$

$$\mathbf{U} = \begin{bmatrix} h \\ hu \\ hv \end{bmatrix}, \mathbf{F}(\mathbf{U}) = \begin{bmatrix} hu \\ hu^2 + \frac{1}{2}gh^2 \\ huv \end{bmatrix}, \mathbf{G}(\mathbf{U}) = \begin{bmatrix} hv \\ huv \\ hv^2 + \frac{1}{2}gh^2 \end{bmatrix}, \mathbf{S}(\mathbf{U}) = \begin{bmatrix} 0 \\ gh\left(S_{0x} - S_{fx}\right) \\ gh\left(S_{0y} - S_{fy}\right) \end{bmatrix}, \tag{4}$$

where $\mathbf{U}$ is the conservative variable, $\mathbf{F}(\mathbf{U})$ and $\mathbf{G}(\mathbf{U})$ are the flow fluxes in the $x$ and $y$ directions, respectively, and $\mathbf{S}(\mathbf{U})$ is the source term, which consists of the bottom slopes $S_{0x}$ and $S_{0y}$ in the $x$ and $y$ directions, respectively, and the friction slopes $S_{fx}$ and $S_{fy}$. In Equation (4), h is the depth, whereas $u$ and $v$ are the velocities in the $x$ and $y$ directions, respectively.

By integrating Equation (3) for an arbitrary computational grid, a basic equation of the finite volume method can be obtained as follows:

$$\int_A \frac{\partial \mathbf{U}}{\partial t} dA + \oint_\Omega \mathbf{E} \cdot \mathbf{n} d\Omega = \int_A \mathbf{S} dA, \tag{5}$$

where $\Omega$ is the boundary of the grid, $d\Omega$ is the length of the interface, $\mathbf{E} = [\mathbf{F}, \mathbf{G}]^T$ is the flow flux vector, and $\mathbf{n}$ is the outward unit normal vector perpendicular to the boundary $\Omega$. Assuming that the conservative variable $\mathbf{U}$ has a constant value in the computational grid, the basic equation of the finite volume method, i.e., Equation (5) can be modified as Equation (6):

$$A\frac{d\mathbf{U}}{dt} + \sum_{m=1}^{M} \mathbf{E}^m L^m = \int_A \mathbf{S}\, dA, \tag{6}$$

where A is the area of the computational grid, $m$ is the exponent representing the grid interface, $L^m$ is the length of the interface m, and $\mathbf{E}^m$ is the flow flux vector computed between the computational grid and the adjacent grid sharing the interface $m$.

### 2.4. Flood Hazard Mapping Using Fuzzy TOPSIS

The flood indices (maximum flood depth, maximum velocity, and maximum flood arrival time) estimated using the 2-D finite volume analysis model were developed based on a nonstructural grid. Each indicator was converted to 5 m × 5 m raster data with the same grid structure for analysis using a GIS tool. The conversion to a raster grid structure enabled comparison of the inundation analysis results obtained from different grid structures and further analysis using GIS tools.

For the levee failure and overtopping analyses, various inundation analysis results were obtained for the same flood index for each grid and scenario combination. Fuzzy logic was introduced to address uncertainties in constructing an integrated flood hazard map.

By sorting the values of each flood index in a raster grid according to frequency, a triangular distribution of values can be obtained, as shown in Figure 3.

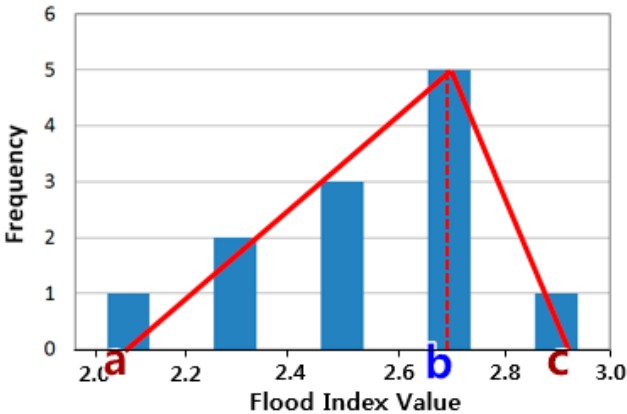

**Figure 3.** Distribution of the index value.

Therefore, as shown in Figure 4, we constructed a triangular fuzzy membership function using the minimum, median, and maximum values (a, b, and c, respectively) of the flood indices calculated in the same raster grid from several scenarios. Among these indices, the highest value was the upper boundary value of the fuzzy number in the triangular fuzzy membership function, the median value was the fuzzy number with a membership function value of 1, and the lowest value was the lower boundary value of the fuzzy number. The lower (a), median (b), and upper (c) layers were constructed in the form of triangular fuzzy functions for all the grids.

### 2.5. Fuzzy TOPSIS Basic Theory

The standard TOPSIS method is a simple ranking method that attempts to choose alternatives that simultaneously have the shortest distance from the positive ideal solution and the farthest distance from the negative-ideal solution [30]. For the supplier selection, Chen et al. [31] proposed a hierarchy multiple criteria decision-making (MCDM) model based on fuzzy-sets theory to deal with the supplier selection problems in the supply chain system.

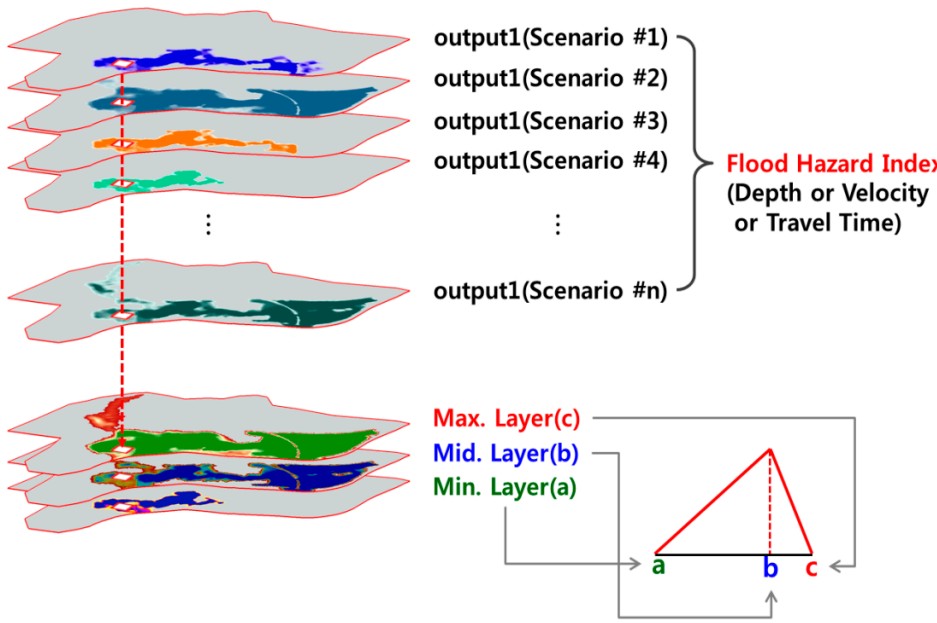

**Figure 4.** Composition of triangular fuzzy membership function.

This study applied a triangular fuzzy membership function to the criteria for TOPSIS. We standardized the membership functions to the same scale while retaining the properties of the triangular fuzzy numbers. The standardized fuzzy decision matrix ($\overline{R}$) is given in Equation (7), where $\overline{r_{ij}}$ denotes the standardized triangular fuzzy numbers:

$$\overline{R} = \begin{bmatrix} \overline{r_{11}} & \overline{r_{12}} & \cdots & \overline{r_{1n}} \\ \overline{r_{21}} & \overline{r_{22}} & \cdots & \overline{r_{2n}} \\ \vdots & \vdots & \ddots & \vdots \\ \overline{r_{m1}} & \overline{r_{m2}} & \cdots & \overline{r_{mn}} \end{bmatrix} \tag{7}$$

This matrix can be standardized using the cost-benefit technique of the triangular fuzzy number $\left( a_{ij}, b_{ij}, c_{ij} \right)$ devised by Chen and Hwang [32]. Equations (8) and (9) are the standardized formulas. In this study, the flood depth and velocity were considered as the benefit criteria, while the maximum flood wave arrival time was considered as the cost criteria.

$$\widetilde{r_{ij}} = \left( \frac{a_{ij}}{c_j^*}, \frac{b_{ij}}{c_j^*}, \frac{c_{ij}}{c_j^*} \right)$$
$$c_j^* = \max c_{ij} \text{(benefit criteria)} \tag{8}$$

$$\widetilde{r_{ij}} = \left( \frac{a_j^-}{c_{ij}}, \frac{a_j^-}{b_{ij}}, \frac{a_j^-}{a_{ij}} \right)$$
$$a_j^- = \min a_{ij} \text{(cos t criteria)} \tag{9}$$

Accordingly, the weight ($\overline{W}$) was applied to the standardized fuzzy matrix ($\overline{R}$) so that the evaluation matrix ($\overline{V}$) could be expressed as Equation (10).

$$\overline{V} = \begin{bmatrix} \overline{v_{11}} & \overline{v_{12}} & \cdots & \overline{v_{1n}} \\ \overline{v_{21}} & \overline{v_{22}} & \cdots & \overline{v_{2n}} \\ \vdots & \vdots & \ddots & \vdots \\ \overline{v_{m1}} & \overline{v_{m2}} & \cdots & \overline{v_{mn}} \end{bmatrix}, \ \widetilde{v_{ij}} = \widetilde{r_{ij}} \times \widetilde{w_{ij}} \tag{10}$$

The fuzzy positive ideal solution (FPIS) and the fuzzy negative ideal solution (FNIS) are given in Equations (11) and (12), respectively. In this paper, the FPIS is the reference value with the highest risk, whereas the FNIS is the reference value with the lowest risk.

$$A^+ = (\widetilde{v_1^+}, \widetilde{v_2^+}, \ldots, \widetilde{v_n^+}) \tag{11}$$

$$A^- = (\widetilde{v_1^-}, \widetilde{v_2^-}, \ldots, \widetilde{v_n^-}) \tag{12}$$

In Equations (11) and (12), $\widetilde{v_1^+} = \max(v_{ij})$ and $\widetilde{v_1^-} = \min(v_{ij})$. The distance between $A^+$ (FPIS), $A^-$ (FNIS), and each substitution variable was obtained by calculating the distance between two triangular fuzzy numbers $\widetilde{a} = (a_1, a_2, a_3)$ and $\widetilde{b} = (b_1, b_2, b_3)$. The spacing between the FPIS and FNIS and the evaluation results of each grid was derived by Equations (13) and (14), and the closeness coefficient of each alternative was derived by Equation (15).

$$d_i^* = \sum_{j=1}^{n} d(\widetilde{v_{ij}}, \widetilde{v_j^*}), \ i = 1, 2, \cdots, m \tag{13}$$

$$d_i^- = \sum_{j=1}^{n} d(\widetilde{v_{ij}}, \widetilde{v_j^-}), \ i = 1, 2, \cdots, m \tag{14}$$

$$CC_i = \frac{d_i^-}{d_i^* + d_i^-} \tag{15}$$

The estimated proximity coefficient value was used as the evaluation value to determine the flood risk. Since the flood depth and velocity were considered as benefit criteria in this study, the area with high priority was the area with the higher risk of flooding. In other words, because the grid with a high proximity coefficient has an evaluation value that is close to the FPIS, this grid was determined to be an area with a high flood risk. Conversely, the grid with a low proximity coefficient has an evaluation value that is close to the FNIS, which indicates an area with a low flood risk.

## 3. Flood Inundation Analysis for Flood Hazard Mapping

### 3.1. Study Area

The city of Gimcheon, South Korea, located in Gyeongsangbuk-do province, was selected as the site for this study. The Gam river and the Jikjisa river pass through this city. These rivers are the first and second tributaries, respectively, of the Nakdong river, which is the longest river in South Korea. Due to the predominantly lowland topography, the study area experiences frequent damage from large typhoons.

### 3.2. Application of 1-D Model

FLDWAV [25], a one-dimensional dynamical hydraulic analysis model, was used to estimate the inflow discharge from the stream to the protected lowland due to levee failure or overtopping. This discharge is subject to uncertainty because of various factors, such as the failure width, time to final failure, and discharge coefficient of the levee. In the event of levee failure and overtopping, the magnitude of the inflow discharge to the protected lowland areas depends on the magnitude of the flood flows in the adjacent rivers. To reflect the uncertainties in levee failure and overtopping, several factors, such as the 100-, 200-, and 500-year frequency flood rates were considered, as well as the levee failure width, failure duration, and overtopping coefficient. Using the HEC-HMS rainfall-runoff analysis model, the design flood hydrograph was applied to the inflow discharge in the main stream and tributaries with reference to available data on the river basin plan [33]. To estimate the levee overtopping and failure discharge that match with the estimated design flood hydrograph, a stream system that consists of the first tributary of the Jikjisa river, as shown in Figure 5, was constructed

and analyzed. The Typhoon Sanba in 2012 was used to calibrate the roughness coefficient, which is the main parameter of the 1-D hydraulic model, and the calibration results are illustrated in Figure 6. The 1-D hydraulic analysis was performed using the upstream and downstream boundary conditions and the lateral inflow amount in the tributary. The upstream boundary conditions and the later inflows were calculated from the HEC-HMS model.

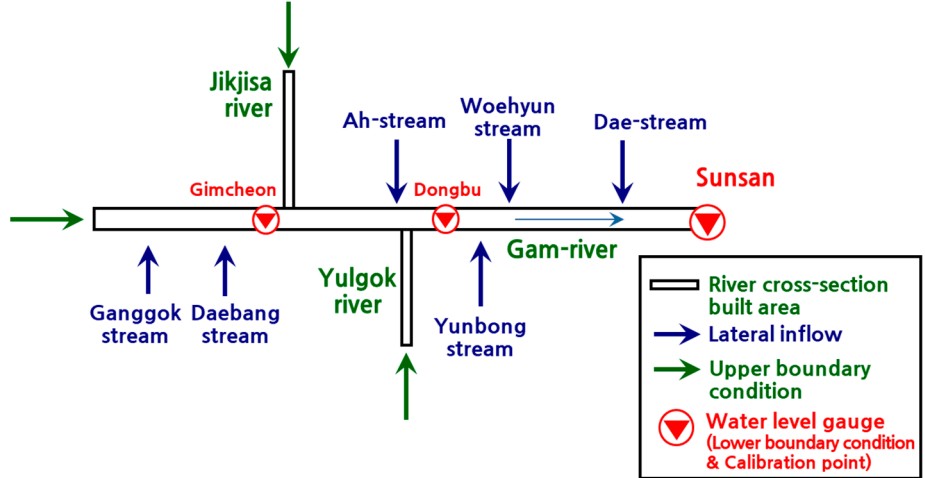

**Figure 5.** 1-D river system of study area.

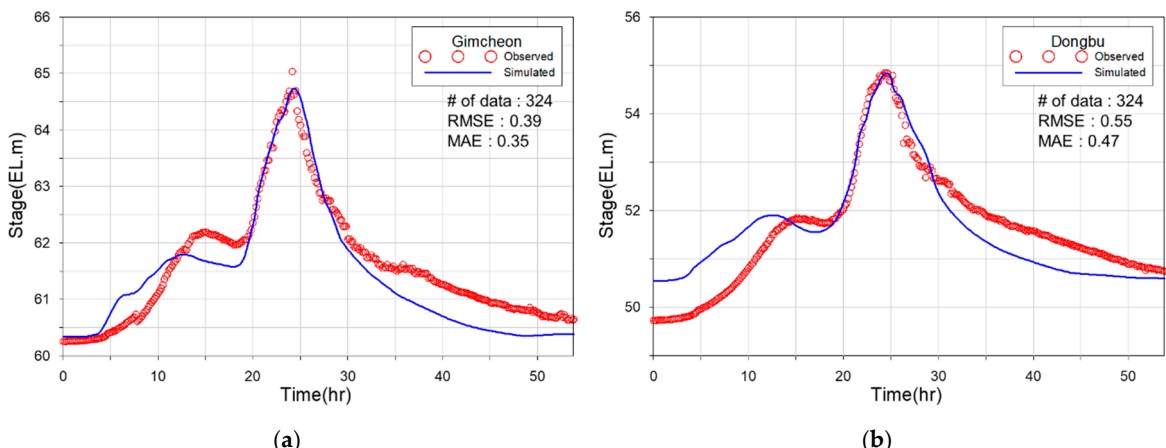

**Figure 6.** Calibration result of 2012 Typhoon Sanba. (**a**) Gimcheon station and (**b**) Dongbu station.

### 3.3. Levee Failure Flow Rate Calculation

To create a flood hazard map for levee failure or overtopping in the Gimcheon city area at the confluence of the Gam and Jikjisa rivers, six levees (Levees #1–6) in the Gam river and three (Levees #7–9) in the Jikjisa river were selected as levee failure and overtopping points. These points were either the locations of previous levee failure or overtopping incidents or those identified as risk points for levee overtopping by the 1-D hydraulic analysis. In particular, the 1-D hydraulic analysis results indicated that the locations of previous overtopping had a high risk of overtopping. Figure 7 shows the selected levee failure or overtopping points. Table 1 lists the information on each levee failure and overtopping point.

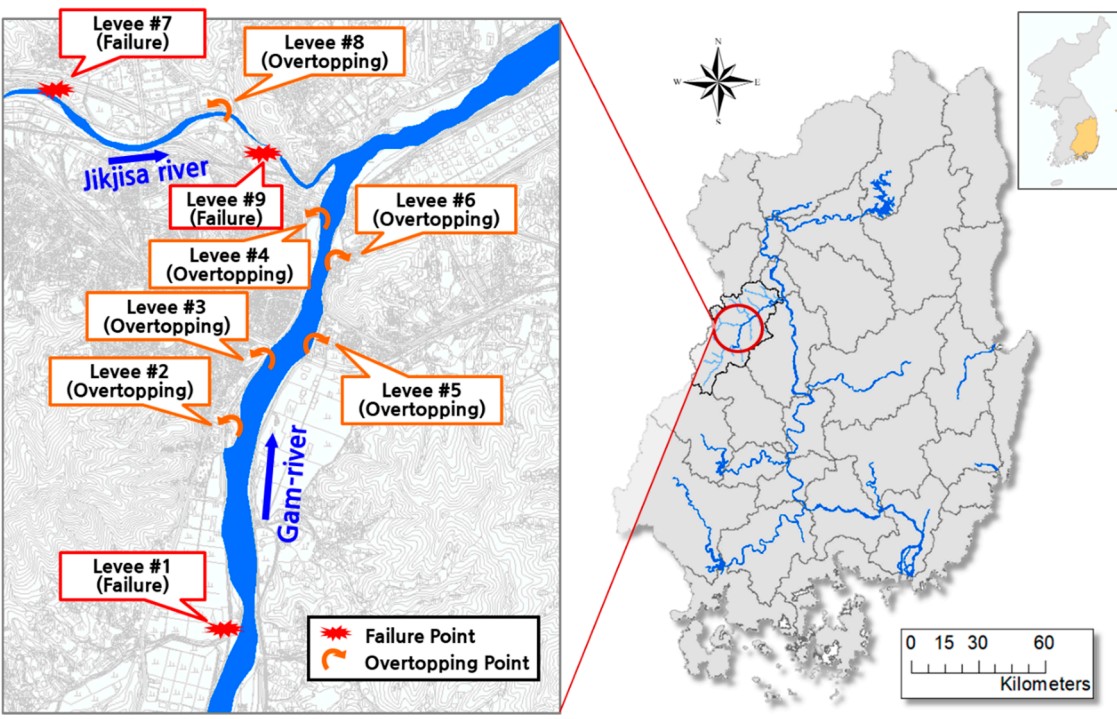

**Figure 7.** Levee failure and overtopping points.

**Table 1.** Information on levee failure and overtopping points.

| Levee No. | Failure/Overtopping | Location | Causes |
|:---:|:---:|:---:|:---:|
| 1 | Failure | Yangok Soha river confluence point | Typhoon Sanba in 2012 |
| 2 | Overtopping | In front of old Yanggeum-dong Community Center | Typhoon Rusa in 2002 and Typhoon Sanba in 2012 |
| 3 | Overtopping | Lowland left of the Gam river railroad bridge | Typhoon Rusa in 2002 |
| 4 | Overtopping | Lowland under Gimcheon Bridge | Risk point by 1-D hydraulic analysis model |
| 5 | Overtopping | Jijwa-dong overtopping point | Typhoon Rusa in 2002 and Typhoon Sanba in 2012 |
| 6 | Overtopping | Baeda-ri overtopping point | Typhoon Rusa in 2002 and Typhoon Sanba in 2012 |
| 7 | Failure | Left of the Jikjisa river railroad bridge | Typhoon Rusa in 2002 |
| 8 | Overtopping | Sineum Greenville point | Typhoon Rusa in 2002 |
| 9 | Failure | Jikjisa river downstream limestone levee | Flood in 1999 |

Three parameters (levee failure width, failure duration, and discharge coefficient) were selected to estimate the inflow discharge to the protected lowland due to levee failure. For Levees #1, 7, and 9, the discharge hydrograph was calculated based on the failure of the three parameters. Figure 8 shows the discharge hydrographs of Levee #1 for the various parameters. This chart indicates that the levee failure peak discharge is most sensitive to the levee failure width, followed by the failure duration and then the discharge coefficient. In particular, the levee failure width had a significantly higher influence on the failure peak discharge than the levee failure duration and discharge coefficient. In addition, another important factor that has a great influence on the levee failure or overflow hydrograph is

rainfall [34]. Hence, in this study, the multiple scenarios were set up for levee failure and overtopping, considering the levee failure width and return period of rainfall. Taking into account the scale and breaching history of the river and levee, the widths of the levee were selected as 50, 150, and 250 m for the Gam river (categorized as a national river), and 50, 100, and 150 m for the Jikjisa river (categorized as a local river). Thus, for the levee failure cases (Levees #1, 7, and 9), nine scenarios were defined in terms of the 100-, 200-, and 500-year frequency floods and the three levee failure widths. For the levee overtopping cases (Levees #2, 3, 4, 5, 6, and 8), three scenarios were defined using only the design flood hydrograph. Figure 9 shows the calculated levee failure and overtopping discharge hydrographs for all the scenarios and these were imposed as the boundary conditions for the 2-D flood inundation analysis. The 2-D models were calibrated to previous flood events from Typhoons Rusa (in 2002) and Sanba (in 2012) in terms of the flooded area and the flood depth [34].

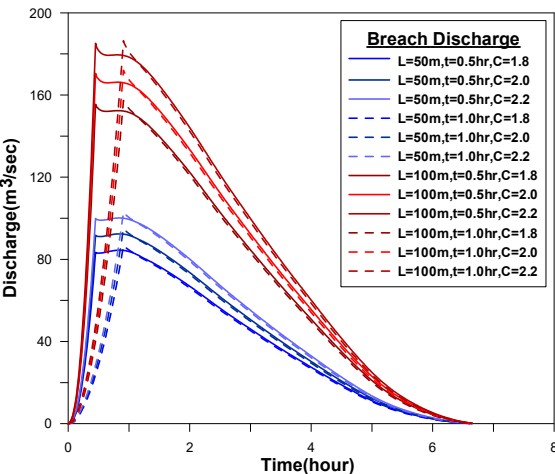

**Figure 8.** Breach discharge hydrographs for different parameter sets at Levee #1 (L: levee failure width, t: failure duration, C: discharge coefficient).

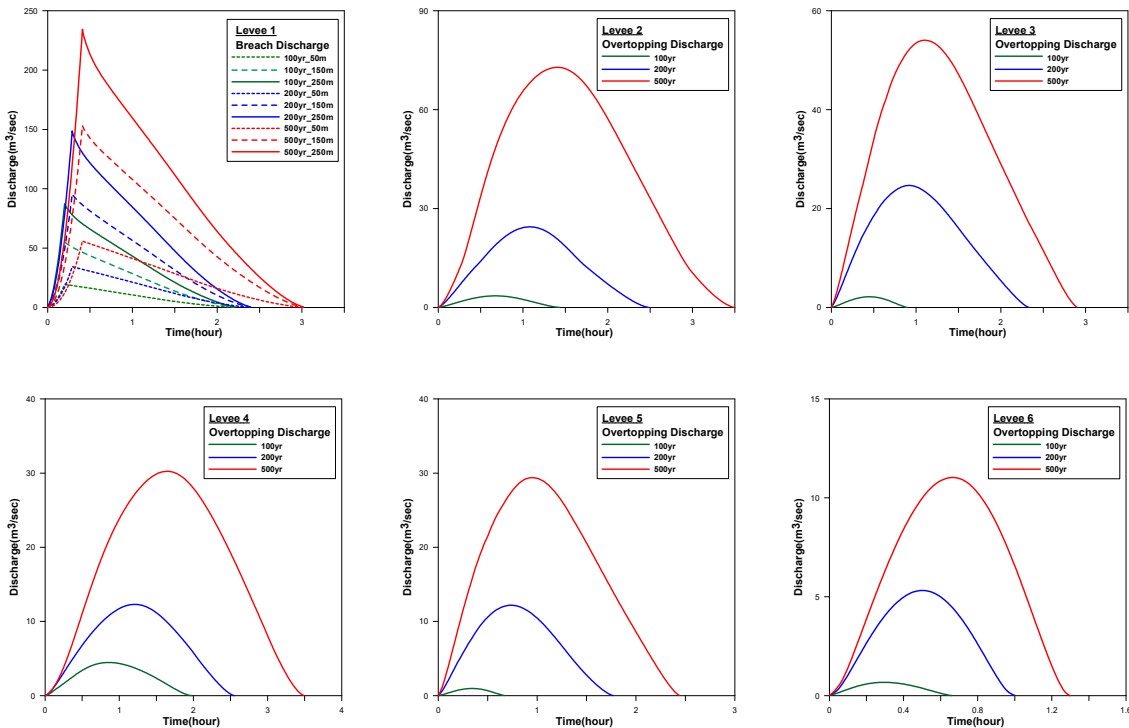

**Figure 9.** *Cont.*

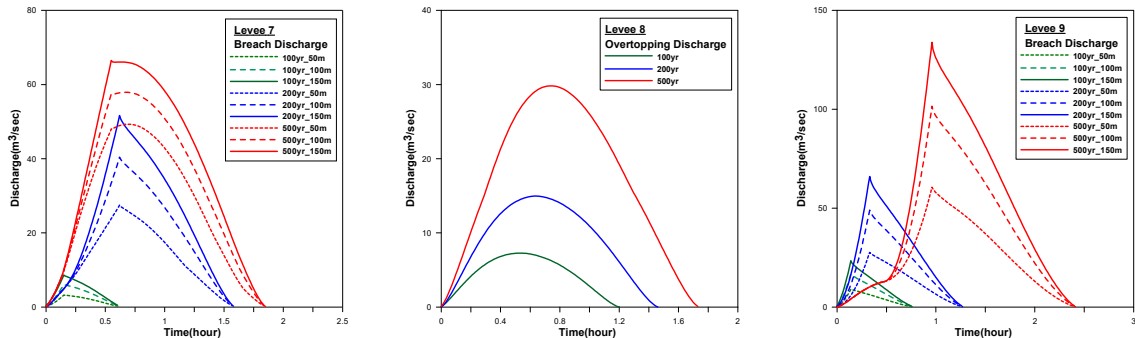

**Figure 9.** Levee failure and overtopping hydrographs for different scenarios.

According to the results of the 2-D inundation analysis, the inundation areas resulting from different levee failure points may overlap. However, in this study, each levee failure was considered independently and the redundancy of two or more levee failures (or inundation analysis results) due to overtopping was excluded. This approach was taken because it was perceived that the scenario uncertainty increases when two or three levee failure cases with overlapping flood areas are considered in addition to the nine levee failure cases. Moreover, since the results of each independent inundation analysis can show the effect of overlapping flood areas in the Fuzzy TOPSIS calculation, the areas where such overlaps occur can be graded as high-risk areas in the flood hazard map.

### 3.4. 2-D Flood Inundation Modeling

An unstructured mesh with a resolution in the range of 5–10 m was generated for the 2-D inundation analysis based on the defined levee failure and overtopping points. A triangulated irregular network (TIN) was created from a 1:1000-scale topographic map and an elevation was assigned to the mesh from the TIN using linear interpolation.

The 2-D inundation analysis was performed using the discharge hydrographs shown in Figure 9 as the upstream boundary conditions. The respective return periods and levee failure widths for these levee failure and overtopping scenarios are given in Table 2. The respective maximum flood depths, maximum velocities, and maximum flood arrival times estimated for the nine levees were considered as independent factors for the construction of the flood hazard map. Figures 10–12 show the results of the 2-D inundation analysis for Levee #1, as an example of the nine levees simulated. Respectively, these figures show the maximum flood depths, maximum velocities, and maximum flood wave arrival times that were calculated for scenarios LV1-1, LV1-6, and LV1-9.

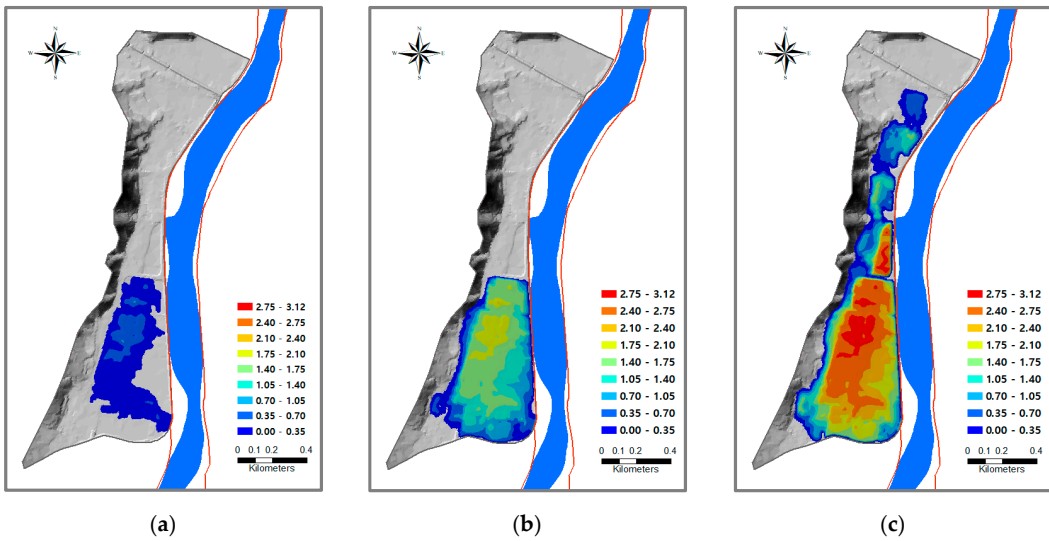

(**a**)    (**b**)    (**c**)

**Figure 10.** Maximum flood depths for Levee #1 failure case, (**a**) LV1-1, (**b**) LV1-6, (**c**) LV1-9.

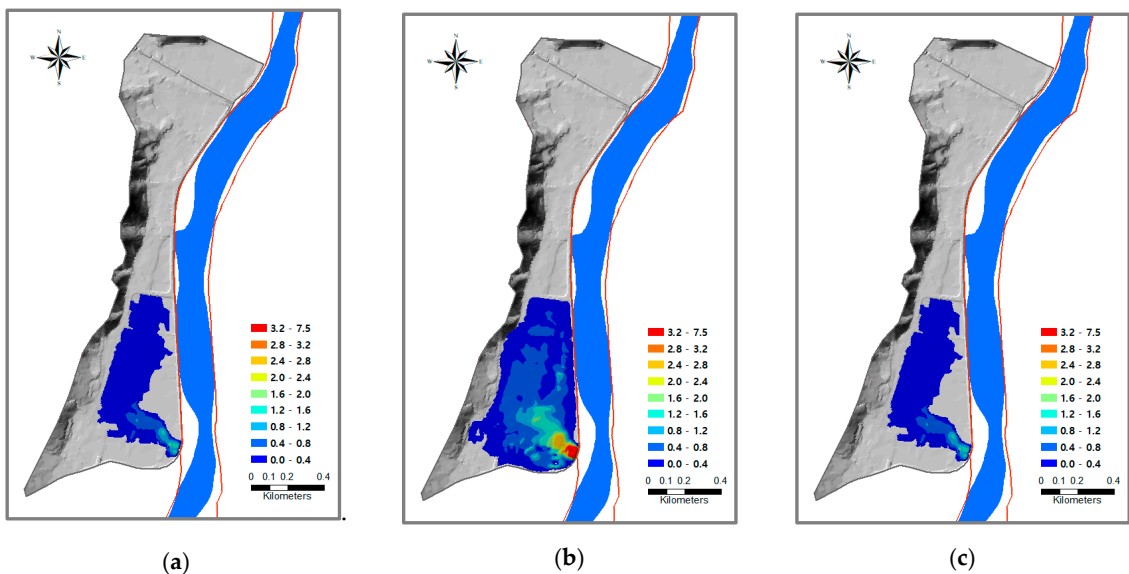

**Figure 11.** Maximum velocities for Levee #1 failure case, (**a**) LV1-1, (**b**) LV1-6, (**c**) LV1-9.

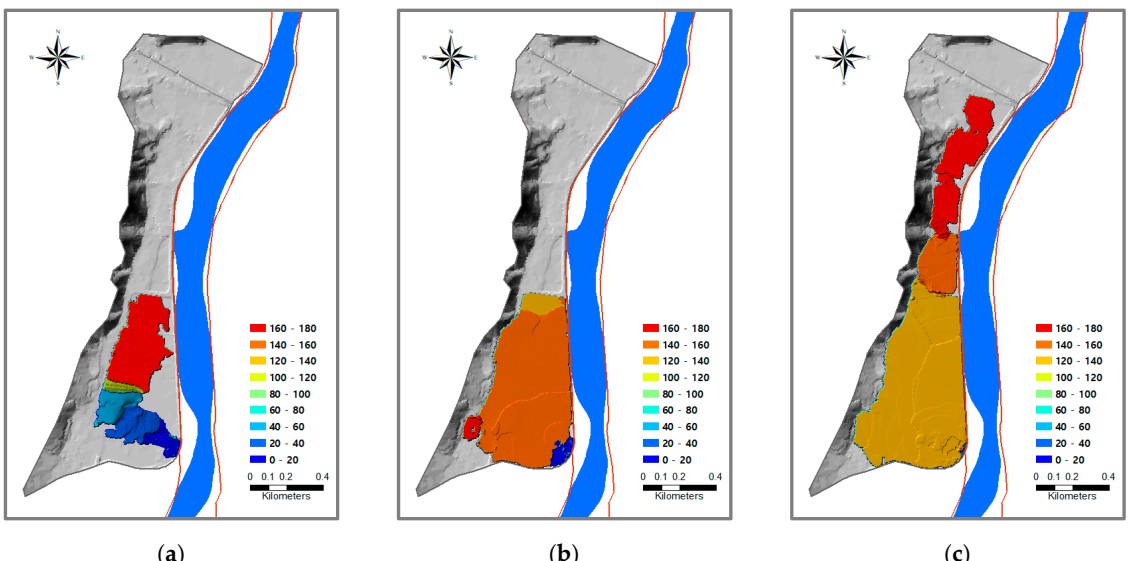

**Figure 12.** Maximum flood wave arrival times for Levee #1 failure case, (**a**) LV1-1, (**b**) LV1-6, (**c**) LV1-9.

**Table 2.** Definition of scenarios with respect to each levee.

| Scenario | Levee No. | Return Period (Year) | Breach Width (m) | Scenario | Levee No. | Return Period (Year) | Breach Width (m) |
|---|---|---|---|---|---|---|---|
| LV1-1 | | 100 | 50 | LV7-1 | | 100 | 50 |
| LV1-2 | | 100 | 150 | LV7-2 | | 100 | 100 |
| LV1-3 | | 100 | 250 | LV7-3 | | 100 | 150 |
| LV1-4 | | 200 | 50 | LV7-4 | | 200 | 50 |
| LV1-5 | 1 (Failure) | 200 | 150 | LV7-5 | 7 (Failure) | 200 | 100 |
| LV1-6 | | 200 | 250 | LV7-6 | | 200 | 150 |
| LV1-7 | | 500 | 50 | LV7-7 | | 500 | 50 |
| LV1-8 | | 500 | 150 | LV7-8 | | 500 | 100 |
| LV1-9 | | 500 | 250 | LV7-9 | | 500 | 150 |
| LV2-1 | | 100 | - | LV8-1 | | 100 | - |
| LV2-2 | 2 (Overtopping) | 200 | - | LV8-2 | 8 (Overtopping) | 200 | - |
| LV2-3 | | 500 | - | LV8-3 | | 500 | - |

**Table 2.** *Cont.*

| Scenario | Levee No. | Return Period (Year) | Breach Width (m) | Scenario | Levee No. | Return Period (Year) | Breach Width (m) |
|---|---|---|---|---|---|---|---|
| LV3-1 | 3 (Overtopping) | 100 | - | LV9-1 | 9 (Failure) | 100 | 50 |
| LV3-2 | | 200 | - | LV9-2 | | 100 | 100 |
| LV3-3 | | 500 | - | LV9-3 | | 100 | 150 |
| LV4-1 | 4 (Overtopping) | 100 | - | LV9-4 | | 200 | 50 |
| LV4-2 | | 200 | - | LV9-5 | | 200 | 100 |
| LV4-3 | | 500 | - | LV9-6 | | 200 | 150 |
| LV5-1 | 5 (Overtopping) | 100 | - | LV9-7 | | 500 | 50 |
| LV5-2 | | 200 | - | LV9-8 | | 500 | 100 |
| LV5-3 | | 500 | - | LV9-9 | | 500 | 150 |
| LV6-1 | 6 (Overtopping) | 100 | - | - | - | - | - |
| LV6-2 | | 200 | - | - | - | - | - |
| LV6-3 | | 500 | - | - | - | - | - |

## 4. Flood Hazard Mapping Using Fuzzy TOPSIS

### 4.1. Fuzzification of Flood Indices

The flood indices (maximum flood depth, maximum velocity, and maximum flood wave arrival time) calculated from the 2-D flood analysis using the nonstructural grid were converted into 5 m × 5 m raster data grids. These data grids were compiled for each levee failure and overtopping scenario listed in Table 2. Subsequently, we used these data grids to compile another set of raster data grids for the (a) lower layer, (b) median layer, and (c) upper layer of the triangular fuzzy membership functions as shown in Figures 3 and 4.

Figure 13 shows the lower, median, and upper layers of the flood indices for the failure of Levee #1. In this figure, the first to third rows of panels represent the triangular fuzzy membership function layers for the maximum flood depth, the maximum velocity, and the maximum flood wave arrival time, respectively.

### 4.2. Selection of Alternative Criteria for Fuzzy TOPSIS

The priority of the relative risk between each raster grid described in Section 4.1 was estimated using the TOPSIS technique. The flood indices (maximum flood depth, maximum velocity, and maximum flood wave arrival time) were used as evaluation criteria for applying the Fuzzy TOPSIS technique. For each criterion, each grid of fuzzified layers (with its triangular fuzzy membership function) was considered as an alternative in the MCDM.

### 4.3. Creating and Grading the Flood Hazard Map

The Fuzzy TOPSIS technique was applied to each flood index such as flood depth, velocity, and arrival time predicted by 2-D inundation analysis. For all the raster grids with flood indices embedded in it, the closeness coefficient was calculated and used to evaluate the degree of flood hazard. The high closeness coefficient represents a high flood hazard. Eventually, the flood hazard maps created in this study are represented by a closeness coefficient which is the final result of the Fuzzy TOPSIS process. The coefficient was used as a measure to represent the degree of flood hazard.

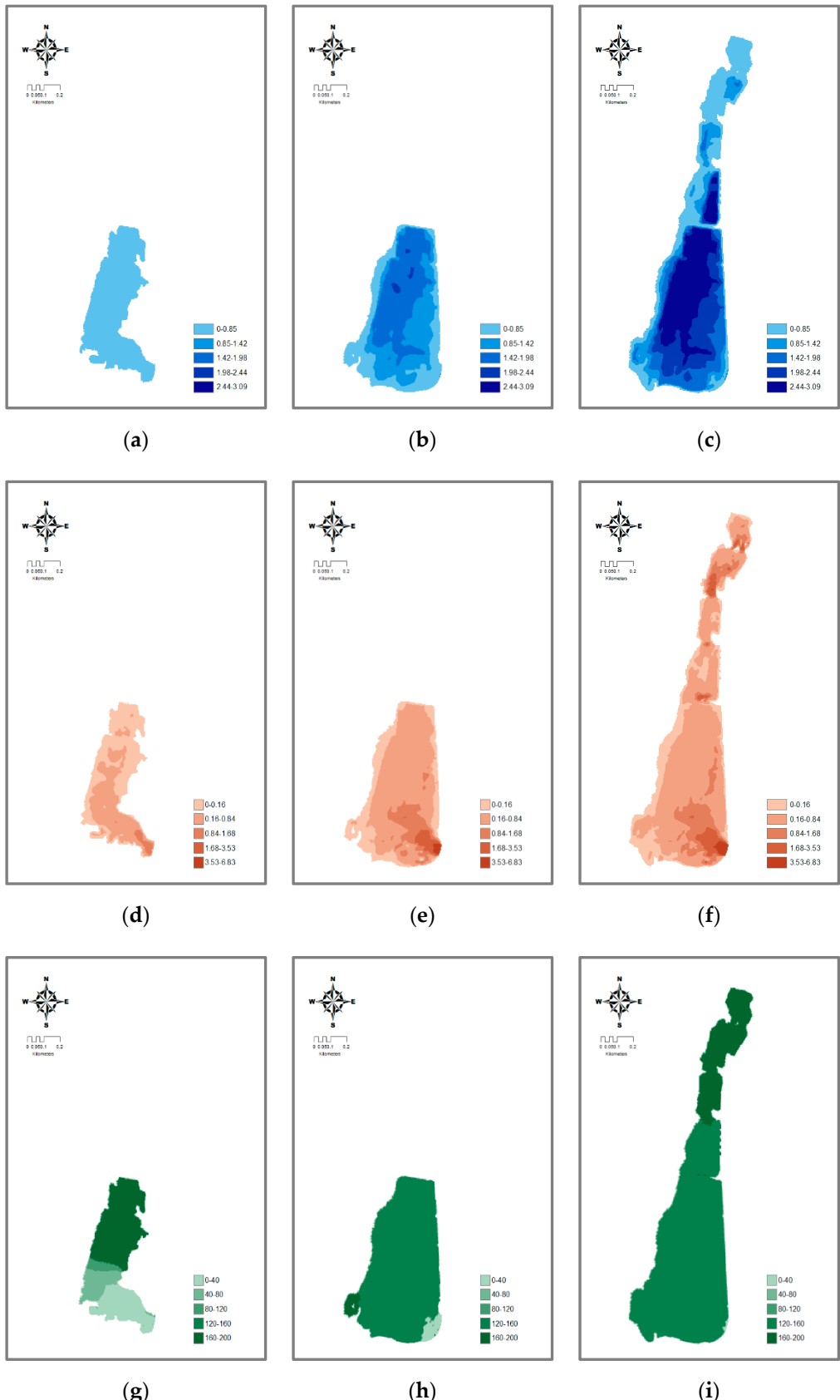

**Figure 13.** Construction of triangular fuzzy numbers of flood indices for Levee #1 failure, (**a**) Lower layer (depth, a), (**b**) Meidan layer (depth, b), (**c**) Upper layer (depth, c), (**d**) Lower layer (velocity, a), (**e**) Meidan layer (velocity, b), (**f**) Upper layer (velocity, c), (**g**) Lower laye r (travel time, a), (**h**) Meidan layer (travel time, b), and (**i**) Upper layer (travel time, c).

To determine the flood hazard grade, the closeness coefficients were standardized and classified into five grades using the Z-score method [35]. As shown in Equation (16), Z-score method is a statistical technique that can be useful for directly comparing statistics between various evaluation indexes with different mean values and standard deviations:

$$Z_i = \frac{X_i - \mu}{\sigma} \tag{16}$$

where, $i$ means raster grids, $Z_i$ is standardized closeness coefficient for each grid. $X_i$ is the value of the closeness coefficient, $\mu$ and $\sigma$ are the mean and standard deviation of closeness coefficient. These calculations from Fuzzy TOPSIS process to flood hazard grading were performed using Python Script within the ArcGIS (ESRI inc., Redlands, CA, USA).

Figure 14a shows the closeness coefficients calculated by the Fuzzy TOPSIS technique for levee failure and Figure 14b shows the corresponding flood hazard grades obtained by the Z-score method. Since the areas inundated by the failure and overtopping of each levee were mapped individually, the areas with high flood hazard grades in Figure 14b would include some areas with overlapping flood impacts.

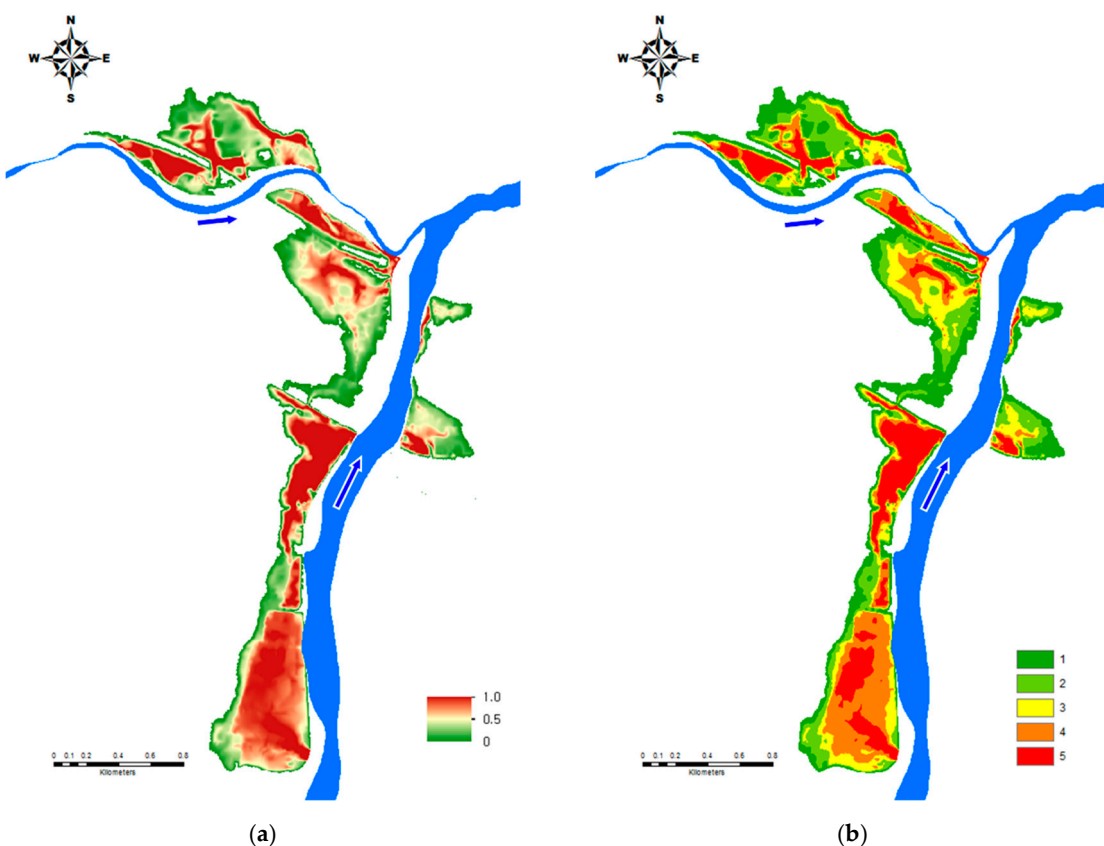

(**a**)　　　　　　　　　　　　　　　　　　　(**b**)

**Figure 14.** Flood hazard map for levee failure. (**a**) Closeness coefficient and (**b**) flood hazard grade.

*4.4. Review of Application Results*

For a detailed analysis, the flood hazard map was overlain on a satellite image, as shown in Figure 15. The highest grade of flood hazards (Grade 5) was identified in seven areas. Table 3 lists the details of each area. Areas 1 and 2 represent rural and urban areas, respectively, and are lowland areas where inundation occurs frequently during major floods, such as Typhoon Rusa (2002) and Typhoon Sanba (2012). In addition, Areas 3 and 4 represent agricultural products market and small residential areas, respectively, and as in Areas 1 and 2, inundation occurs frequently in these areas during major floods. Area 5 is situated near the Gimcheon Girls' High School, where flood damage was reported

to have occurred during Typhoon Rusa in 2002. Although inundation did not occur during Typhoon Sanba in 2012, this area has a high flood risk because of its lowland topography. Area 6 covers the location of the supermarket where underground spaces were flooded during Typhoon Rusa in 2002. Area 7 covers the sculpture park on the riverside, which was flooded entirely during Typhoon Rusa in 2002 and Typhoon Sanba in 2012.

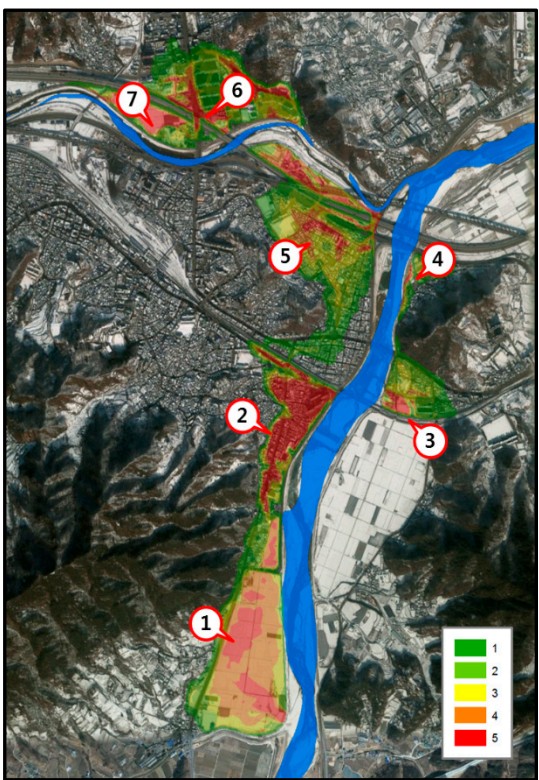

**Figure 15.** Flood hazard map of this study.

**Table 3.** Comparison of flood hazard map between this study and MOLIT in the real flood areas.

| No. | Site | This Study (Figure 15) Grade | MOLIT [33] (Figure 16) Depth (m) | Remarks |
|---|---|---|---|---|
| 1 | Rural area | Grade 5 | 2.0~5.0 | |
| 2 | Lowland residential area | Grade 5 | 2.0~5.0 | Flood prone area |
| 3 | Agricultural products market area | Grade 5 | 2.0~5.0 | |
| 4 | Small-scale residential area | Grade 5 | Non flood | |
| 5 | Gimcheon girls' high school | Grade 5 | 2.0~5.0 | Flooded during Typhoon Rusa |
| 6 | Supermarket area | Grade 5 | Non flood | Underground space flooded during Typhoon Rusa |
| 7 | Sculpture park | Grade 5 | Non flood | Flooded during Typhoon Rusa and Typhoon Sanba |

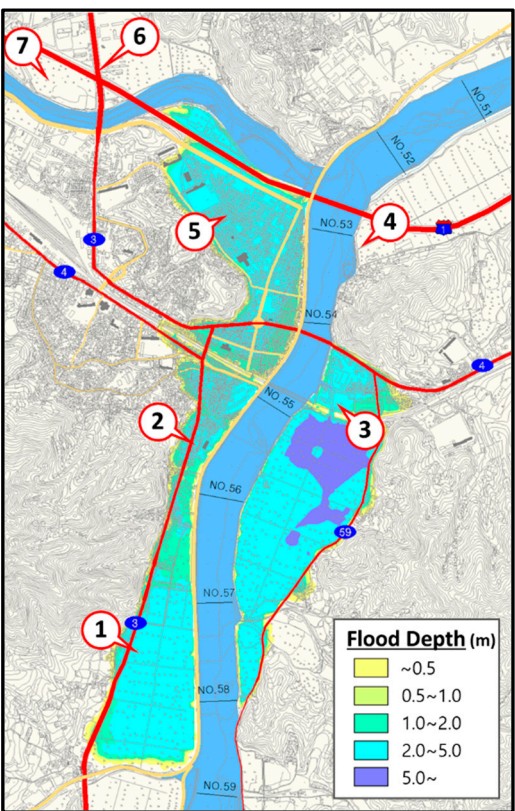

**Figure 16.** Flood hazard map of Ministry of Land, Infrastructure and Transports (MOLIT) [33].

Figure 16 shows the flood hazard map for the Gam river that was created by the current flood hazard mapping method from Ministry of Land, Infrastructure and Transports (MOLIT) [33]. Table 3 presents a comparison of flood inundation maps created in this study and MOLIT [33] for areas where actual floods occurred. The highest grade (Grade 5) areas estimated in this study were well matched with areas where actual floods occur frequently. On the other hand, flood hazard map of MOLIT showed that areas of No. 1, 2, 3, and 5 were more than 2.0 m in flood depth, and areas No. 4, 6, and 7 were not flooded.

The differences in the flood extents between the two maps are explained for the following reasons. The MOLIT [33] flood hazard map was based on the arbitrarily chosen locations according to the flow characteristics such as the past levee failure history or the confluence of the small rivers as a scenario (e.g., the lower area of No. 3 and the No. 4 area), unlike this study in which a 2-D inundation analysis was performed by creating the failure or overtopping scenarios of vulnerable levees based on the hydraulic analysis results. Furthermore, the MOLIT [33] flood hazard map was constructed by considering only the levee failure of the national river (the Gam river). The flooding of areas No. 6 and 7 in the flood hazard map produced in this present study was not considered in the MOLIT [33] flood hazard because the Jikjisa river is not managed by the MOLIT [33].

The flood hazard map of MOLIT [33] expresses the hazardous area by flooding extent and depth, while the flood hazard map of this study is graded using the closeness coefficient, which was based on the flood indices (flood depth, velocity, and arrival time). In addition, this study fuzzified the flood indices predicted from scenario-based 2-D flood models in order to consider uncertainties in models depending on varying circumstances such as rainfall frequency and levee failure shape. Then, Fuzzy TOPSIS was applied to integrate each flood index and represent it as a graded, single flood hazard map. It is expected to provide policy makers with more objective information in decision making considering flood risk by producing a graded, integrated flood hazard map from the methodology proposed in this study. Though, this study, which proposed a method to rank flood hazard by applying

closeness coefficient, needs further study to intuitively quantify the flood hazard, such as flood depth and velocity.

## 5. Conclusions

This study aims to improve the methodology that is currently used to construct the flood hazard map in South Korea. Currently, several flood hazard maps are produced for each flood frequency by overlaying the maximum flood depth and extent derived from flood inundation analysis and this methodology does not consider the uncertainties of flood indices (flood depth and extent). This paper has proposed a more advanced methodology for constructing flood hazard maps and this methodology was applied to the Gam river in South Korea. The flood hazard map was created by considering levee failure and overtopping at the points of past levee failures near Gimcheon City. Various results were obtained based on different scenarios using a 2-D inundation analysis model. The results of this model were used as flood indices (flood depth, velocity, and arrival time) for producing the flood hazard map. The main conclusions drawn from this study are as follows:

(1) A fuzzy-based theory was applied in creating a flood hazard map by considering the uncertainties in a 2-D inundation analysis. This study directly considered the uncertainties in the flood indices using triangular fuzzy membership functions by applying various scenarios to improve the current methodology that does not consider the uncertainties of flood indices. The results of this study presented that the Fuzzy MCDM technique can be used to produce flood hazard maps.

(2) By comparing the flood hazard map produced in this study with the current flood hazard map of the Gam river, the methodology proposed in this study was found to be more advantageous than the current method with regard to the accuracy in the comparison to actual flood prone areas, the grading of the inundated areas, and an integrated single map considering various flood scenarios.

(3) In this study, the flood hazard map was generated using only the inundation analysis results due to levee failure and overtopping. Nevertheless, the methodology proposed in this study is expected to expand to inundation analysis and integrated flood hazard mapping in the case of other floods, including dam failure and urban flood.

**Author Contributions:** All authors contributed extensively to the work. B.K. and K.-Y.H. conceptualized and designed the study. T.H.K. produced the data required to apply the methodology to the study area and conducted the model simulation, validation and writing-original draft preparation. B.K. analyzed the results, completed the manuscript and wrote review and editing. K.-Y.H. supervised the project and acquired the funding.

**Funding:** This research was funded by the Korea Ministry of Environment (MOE).

**Acknowledgments:** This subject is supported by Korea Ministry of Environment (MOE) as "Water Management Research Program" (RE201901062).

**Conflicts of Interest:** The authors declare no conflict of interest.

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
