# Peer review of "Application of Fuzzy TOPSIS to Flood Hazard Mapping for Levee Failure"

_water, doi:10.3390/w11030592_

Round 1

Reviewer 1 Report

The evaluated manuscript “Application of Fuzzy TOPSIS to Flood Hazard Mapping for Levee Failure” written by Tae Hyung Kim, Byung-Hyun Kim and Kun-Yeun Han deals with very important and very current problem of flood hazard analysis under pressure of different and uncertain factors. The Authors focus on flood hazard related to dike failure. The basis for the analysis are simulations of 2D hydrodynamic model developed by the Authors and described in earlier papers. The fuzzy-based TOPSIS model is proposed as the main method for construction of flood hazard maps. The main idea is to include uncertainty related to failure location and three parameters describing the characteristics of the failure: levee failure width, failure duration, and discharge coefficient.

The problem analyzed is very complex. The approach presented is original and well developed. The description of the problem and applied methodology seems to be good enough. The conclusions are  There are some corrections which may be introduced to the text of the manuscript, but these elements do not decrease the overall value of the paper. The suggested corrections are listed below.

The only element which may raise some concern is practical application of the presented methodology. In my opinion the Authors should consider stakeholders expectations and abilities to absorb such kind of information which is given in fuzzy flood hazard maps. Maybe the complexity of the proposed methodology should be reduced. It would good if the Authors add some comments in this in the conclusions.

Suggested corrections:

Lines 41-43

Supposedly the Authors mean "more accurate", not "more   efficient". The 1D models are more efficient, because 1D simulation is   faster than 2D. Additionally the accuracy of 1D map is similar to those   produced from 2D results in many cases.

Lines 52-53

It's not clear what the Authors mean. The standard approach is to   choose proper scenario based on the analysis of maximum flows. The maximum   flow means the flow with assigned probability of exceedance or return time. It   may be applied as the culmination of the design flood wave composed on the   basis of historical events.

What kind of scenarios the Authors mean?

Lines 81-82

The sentence is not exactly true. There are also flood hazard maps   presenting spatial distribution of the velocity magnitudes and directions.

Subsection 2.1

It's worth to emphasize that the model was developed by the Authors.

Additionally the description of the model should be completed with   description of initial and boundary conditions applied.

Lines 119-120

There should be provided some measure of mesh quality, e.g. max and   mean cell size, etc..

Lines 213-215

What kind of parameters were calibrated? Roughness coefficients?

Figure 5

It should be good to provide some quantitative measures of calibration   fit, e.g. mean square error, etc..

entire text

The term “dam failure” is used. In my opinion “dike failure” is fits   better what is really analyzed, but I’m not Native Speaker. Please, check   correctness of this term.

Author Response

Application of Fuzzy TOPSIS to Flood Hazard Mapping for Levee Failure

By Kim, Tae Hyung, Kim, Byung-Hyun and Han, Kun-Yeun

Point by Point Response to Reviewer Comments

The evaluated manuscript “Application of Fuzzy TOPSIS to Flood Hazard Mapping for Levee Failure” written by Tae Hyung Kim, Byung-Hyun Kim and Kun-Yeun Han deals with very important and very current problem of flood hazard analysis under pressure of different and uncertain factors. The Authors focus on flood hazard related to dike failure. The basis for the analysis are simulations of 2D hydrodynamic model developed by the Authors and described in earlier papers. The fuzzy-based TOPSIS model is proposed as the main method for construction of flood hazard maps. The main idea is to include uncertainty related to failure location and three parameters describing the characteristics of the failure: levee failure width, failure duration, and discharge coefficient.

The problem analyzed is very complex. The approach presented is original and well developed. The description of the problem and applied methodology seems to be good enough. The conclusions are There are some corrections which may be introduced to the text of the manuscript, but these elements do not decrease the overall value of the paper. The suggested corrections are listed below.

The authors thank the reviewer for recognizing the importance of this topic and recommending areas for improvement. Author Comments appear in BLUE, while Reviewer Comments appear in BLACK.

The only element which may raise some concern is practical application of the presented methodology. In my opinion the Authors should consider stakeholders expectations and abilities to absorb such kind of information which is given in fuzzy flood hazard maps. Maybe the complexity of the proposed methodology should be reduced. It would good if the Authors add some comments in this in the conclusions.

The authors have made substantial revisions to the paper. Introduction has been largely revised and Section 2.1 has been added to the manuscript to improve the motivation, objective and methodology of this study. The proposed method of this study can be seen to be somewhat complicated because it involves several processes for flood hazard mapping. Thus, the authors have added “Figure 1 flow chart of this study” to Section 2.1 to help readers easily understand the process. In addition, as the reviewer’s suggestion, we have mentioned the utilization and expectation of this study in Section 4.4 and Conclusion as following.

“It is expected to provide policy makers with more objective information in decision making considering flood risk by producing a graded, integrated flood hazard map from the methodology proposed in this study.”

.

Comments:

Lines 41-43: Supposedly the Authors mean "more accurate", not "more efficient". The 1D models are more efficient, because 1D simulation is faster than 2D. Additionally the accuracy of 1D map is similar to those produced from 2D results in many cases.

"more effcient" has been replaced by "more useful".

A 2-D model is able to provide more spatially detailed information including flood depth and velocity than a 1-D model, so we can get more specific and useful information from flood hazard maps applied 2-D model. Thus, “useful” is used instead of “efficient”

Lines 52-53: It's not clear what the Authors mean. The standard approach is to choose proper scenario based on the analysis of maximum flows. The maximum flow means the flow with assigned probability of exceedance or return time. It may be applied as the culmination of the design flood wave composed on the basis of historical events. What kind of scenarios the Authors mean?

Numerical models have limitations to reflect real-world situations, and these limitations cause model uncertainty. Scenarios that take into account various situations are considered to reflect these uncertainties in numerical simulations. As the reviewer pointed out, the meaning of the sentence was ambiguous, and the contents of the paragraph containing the sentence were revised as follows.

Flood damage can vary in size and range depending on the amount of rainfall, the failure shape of hydraulic structures, and the duration of the failure. Therefore, when carrying out a 2-D inundation analysis for flood hazard mapping, various scenarios should be considered to reflect these uncertainties. Currently, the flood hazard maps being created in Korea do not consider the uncertainties of multiple flood scenarios. There have been many studies on risk analysis, reliability analysis, probabilistic methods including Monte-Carlo simulation to reflect these uncertainties in flood hazard mapping [7, 8, 9]. This study suggests a methodology for flood hazard mapping using Fuzzy MCDM (Multiple Criteria Decision Making) techniques among various methods.”

Lines 81-82: The sentence is not exactly true. There are also flood hazard maps presenting spatial distribution of the velocity magnitudes and directions.

Thanks for the correction. The flood velocity as well as flood depth have been added to this sentence

Subsection 2.1: It's worth to emphasize that the model was developed by the Authors. Additionally the description of the model should be completed with description of initial and boundary conditions applied.

The authors added the phrase (“a 2-D high-accuracy finite volume model developed by co-authors of this study [27, 28]”) to Section. 2.3.

We have made substantial revisions to the paper. The authors have added the Sections “2.1 method of this study” and “2.2 HEC-HMS and FLDWAV” to describe in detail the methodology and models used in this study. Since we cannot allocate lots space to the 2-D model in Section 2.3, the description of boundary conditions have been replaced with references [27, 28].

Lines 119-120: There should be provided some measure of mesh quality, e.g. max and mean cell size, etc.

The size range of unstructured meshes applied for 2-D flood inundation analysis is described in Section 3.4

Lines 213-215: What kinds of parameters were calibrated? Roughness coefficients?

Calibration was performed on the roughness coefficient, which is the main parameter of 1-D hydraulic model. In the manuscript, we have clarified the meaning by revising the sentence as follows.

“The Typhoon Sanba in 2002 was used to calibrate the roughness coefficient, which is the main parameter of the 1-D hydraulic model, and the calibration results are illustrated in Figure 6.”

Figure 5: It should be good to provide some quantitative measures of calibration fit, e.g. mean square error, etc.

The authors added the numbers of data as well as RMSE (Root Mean Square Error), MAE (Mean Absolute Error) between measurements and computations to Figure 6.

Entire text: The term “dam failure” is used. In my opinion “dike failure” is fits better what is really analyzed, but I’m not Native Speaker. Please, check correctness of this term.

The "dam failure" mentioned in the manuscript was used in a different sense from the “levee failure” or “dike failure” used for the flood hazard mapping in this study. The meaning is like that “the methodology proposed in this study can be applied equally to dam failure as well as levee (dike) failure.” Thus, "dam failure" rather than "dike failure" is better suited to the authors' intentions.

The authors thank for the reviewer for carefully reviewing the paper and providing and assessment

Reviewer 2 Report

Referee report on manuscript „Application of Fuzzy TOPSIS to Flood Hazard Mapping for Levee Failure”

The manuscript describes an alternative method for re-classifying inundation maps and producing hazard zoning maps. The presented approach is based on the TOPSIS approach, extended with a fuzzy logic classification.

The manuscript is written in form of a project report. The presented content is interesting. However, the manuscript does not fulfill the criteria to be published as a scientific paper. Thus, I recommend to reject the paper. This recommendation is based on the following main remarks:

-First of all, the manuscript does not have a research question explicitly stated.

-Second, the authors conclude in the Conclusions chapter that the presented new method is “more advantageous than the existing methods”. However, the authors do not prove this statement with transparent criterias or quantitative arguments.

-Third, The first sentence with which the manuscript begins states that flood hazard maps are reducing flood damage. I am really wondering how this works. In my experience, hazard maps avoid an increase of damage potential but they do not reduce damage.

-Furthermore, the added value of the fuzzy logic approach is not demonstrated at all.

Minor remarks:

-The authors do not define the term “hazard maps”. A reader must know what purpose they have in Korea.

-line 42: how “efficient” is defined? Which criterias are used to measure efficiency?

-the TOPSIS method has to be explained in more detail (in words).

-I do not understand why the maximum of the flood wave arrival time is the criteria and not the minimum. Should not short arrival times be classified as more dangerous than slow ones?

Author Response

Application of Fuzzy TOPSIS to Flood Hazard Mapping for Levee Failure

By Kim, Tae Hyung, Kim, Byung-Hyun and Han, Kun-Yeun

Point by Point Response to Reviewer Comments

Referee report on manuscript „Application of Fuzzy TOPSIS to Flood Hazard Mapping for Levee Failure”

The manuscript describes an alternative method for re-classifying inundation maps and producing hazard zoning maps. The presented approach is based on the TOPSIS approach, extended with a fuzzy logic classification.

The manuscript is written in form of a project report. The presented content is interesting. However, the manuscript does not fulfill the criteria to be published as a scientific paper. Thus, I recommend to reject the paper. This recommendation is based on the following main remarks:

The authors thank the reviewer for carefully reviewing the paper and providing a critique. The reviewer’s comments had a major impact one the revision of the paper. We have made substantial revisions to the paper as requested and we are confident that this paper will have a positive impact on the flood risk and flood hazard mapping communities. The contents of Abstract, Introduction, Section 2, Section 4 and Conclusion have been substantially revised.

The authors also thank the reviewer for recognizing the importance of this topic and recommending areas for improvement. Author Comments appear in BLUE, while Reviewer Comments appear in BLACK.

Comments:

- First of all, the manuscript does not have a research question explicitly stated.

The authors have largely revised the Introduction and added Section 2.1 to improve the motivation, objective and methodology of this study. The research question of this study is to reflect the uncertainties for flood hazard mapping using numerical model. However, the traditional methods currently used in Korea for flood hazard mapping are not taking into account these uncertainties. Therefore, this study proposes a new approach for integrating TOPSIS method and fuzzy logic to consider uncertainties in more rational way in the scenario-based flood hazard mapping using 2-D inundation model

- Second, the authors conclude in the Conclusions chapter that the presented new method is “more advantageous than the existing methods”. However, the authors do not prove this statement with transparent criterias or quantitative arguments.

The authors have largely revised the Section 4.4 to describe the differences between the method of this study and currently applied method in Korea for flood hazard mapping by the flood hazard maps created from two methods.

- Third, The first sentence with which the manuscript begins states that flood hazard maps are reducing flood damage. I am really wondering how this works. In my experience, hazard maps avoid an increase of damage potential but they do not reduce damage.

We agree that the reviewer’s comment. The sentence has been revised as reviewer’s suggestion.

-Furthermore, the added value of the fuzzy logic approach is not demonstrated at all.

This study introduced fuzzy logic to take into account the uncertainty of flood indices (flood depth, velocity and arrival time), and the flood indices calculated by various scenarios were fuzzified using the triangular fuzzy membership function. The fuzzy process for flood hazard mapping is detailed in Section 2.4. The accuracy of proposed methodology was presented in revised Section 4.4 by comparing the flood hazard map with the area where actual floods occur frequently. The added value of this study is that the methodology proposed is able to be applied to other floods including dam failure or urban flooding.  

Minor remarks:

-The authors do not define the term “hazard maps”. A reader must know what purpose they have in Korea.

We have added the definition and objective of hazard map to introduction section as reviewer’s suggestion.

-line 42: how “efficient” is defined? Which criterias are used to measure efficiency?

"more effcient" has been replaced by "more useful".

A 2-D model is able to provide more spatially detailed information including flood depth and velocity than a 1-D model, so we can get more specific and useful information from flood hazard maps applied 2-D model. Thus, “useful” is used instead of “efficient”

-the TOPSIS method has to be explained in more detail (in words).

The authors have added the description of TOPSIS method and Fuzzy TOPSIS technique in Section 2.4. In addition, Section 2.1 “method of this study” has been added to describe in detail the methodology of this study. The description and equation on the closeness coefficients and Z-score method also have been added in Section 4.3

-I do not understand why the maximum of the flood wave arrival time is the criteria and not the minimum. Should not short arrival times be classified as more dangerous than slow ones?

Maximum flood arrival time means the arrival time of maximum water level. As reviewer’s comment, the shorter the arrival time, the higher the risk. Fuzzy TOPSIS is considering the factors that have higher risk if the value is larger (e.g. depth, velocity) are considered benefit criteria, conversely the factors that have higher risk if the value is smaller (e.g. arrival time) are considered cost criteria. These are included in the descriptions of the Eq. (8) and (9) in Section 2.4. To clarify the meaning in Section 2.4, the sentence has been revised as follow:

“In this study, the flood depth and velocity (the factors that have higher risk if the value is larger) were considered as the benefit criteria, while the maximum flood wave arrival time (the factors that have higher risk if the value is smaller) was considered as the cost criterion.”

The authors thank for the reviewer for carefully reviewing the paper and providing and assessment

Author Response

Application of Fuzzy TOPSIS to Flood Hazard Mapping for Levee Failure

By Kim, Tae Hyung, Kim, Byung-Hyun and Han, Kun-Yeun

Point by Point Response to Reviewer Comments

Paper Review: Application of Fuzzy TOPSIS to Flood Hazard Mapping for Levee Failure This paper presents a new flood hazard mapping technique using three parameters. The trial to demonstrate the technique is fine, however, there are many things that should be improved. Please consider many comments as follows:

The author’s appreciate the reviewer’s detailed evaluation. The reviewer feedback was extensive and gave us many ideas for improving the paper. The authors also thank the reviewer for recognizing the importance of this topic and recommending areas for improvement. Author Comments appear in BLUE, while Reviewer Comments appear in BLACK.

Minor Comments:

1. The authors choose FLDWAV as 1-D hydraulic analysis model. However, the description about FLDWAV model is insufficient. Authors should supplement a description of the model used in the study. Also, the details on calibration and verification were needed to be presented. The manuscript just highlights The 2012 typhoon Sanba event was used to calibrate the parameters of the 1-D hydraulic model (line 206) and A parametric calibration and verification was performed (line 213). Authors need to describe more details on the process of calibration and parameters of 1-D hydraulic model used to calibrate?

The authors have added Section 2.2 “HEC-HMS and FLDWAV” to describe on models used in this study. Calibration was performed on the roughness coefficient, which is the main parameter of 1-D hydraulic model. In the manuscript, we have clarified the meaning by revising the sentence. In addition, the numbers of data as well as RMSE (Root Mean Square Error), MAE (Mean Absolute Error) between measurements and computations have been added to Figure 6.

2. The major comment that I have concerns the description of the methodology used in this study. The authors should have presented more details on the methodology, its implementation. For example, in chapter 4.3, authors need to present more details on the closeness coefficients and Zscore method used to determine the flood hazard grade. Also, where is the details of the HECHMS modelling?

The authors have added the Sections “2.1 method of this study” and “2.2 HEC-HMS and FLDWAV” to describe in detail the methodology and models used in this study. We also have added the description and equation on the closeness coefficients and Z-score method in Section 4.3 as reviewer’s suggestion. In addition, we have added the description on the TOPSIS method and Fuzzy TOPSIS technique in Section 2.3

3. In introduction, authors should add references to support your descriptions and arguments. For example, you explained about the flood hazard maps in the United States. Where did you get the information? Please provide proper references.

The authors have expanded the citation of related work in the introduction to improve the motivation of this study

4. In the chapter 3.2, I suggest that please provide detailed background. For example, what is the FLDWAV? Readers cannot find the background and details. How about the HEC-HMS modelling? Also, authors calibrated the 1-D hydraulic model using 2012 Typhoon Sanba. I doubt whether the calibrated parameters can represent proper hydraulic process simulation or not as a single event is not enough to calibrate the model

The authors have added to Section “2.1 method of this study” for description detail of methodology of this study. We have added a study flow chart to help readers easily understand this study. In addition, we has added to Section “2.2 HEC-HMS and FLDWAV” to describe on models used in this study. This Section contains the background theory on HEC-HMS and FLDWAV. In general, the hydrodynamic model (e.g., 1-D FLDWAV, 2-D Finite volume model) requires calibration only with roughness coefficient, unlike hydrological models with many parameters that require calibration. Therefore, the hydrodynamic model uses mainly one flood event in order to calibrate the roughness. For this reason, this study also used one flood event (Typhoon Sanba in 2012).

5. In the manuscript, there are many sentences of English which should be rewritten.

Minor Comments:

1. [Abstract] The abstract should include brief descriptions of the purpose, method, and results of the study, but this abstract presented the method description too much compared to the results. Please describe the summary of the results in the abstract.

The authors have revised a lot the abstract as suggested by the reviewer.

2. [Abstract] Korea -> South Korea

“Korea” has been replaced by “South Korea”

3. [Abstract] such as -> e.g.

The sentence including “such as” has been dropped

4. [Abstract] What is the existing maps? Are you talking the convention maps which is developed previously?

“existing” has been replaced by “current”. This study aims to improve the methodology which is currently used to construct the flood hazard map in South Korea

5. [Introduction] The last paragraph of the introduction lacks a description of the purpose and motivation of the study. Authors should describe the motivation of the study instead of overview and traditional methodology in the last paragraph of the introduction.

The authors have largely revised the introduction and added section 2.1 to improve the motivation, objective and methodology of this study. The motivation and objective of this study are to reflect the uncertainties for flood hazard mapping using numerical model. However, the traditional methods currently used in Korea for flood hazard mapping are not taking into account these uncertainties. Therefore, this study proposes a new approach for integrating TOPSIS method and fuzzy logic to consider uncertainties in more rational way in the scenario-based flood hazard mapping using 2-D inundation model

6. [Introduction] Please introduce and describe the paper structure at the end of the introduction section.

The authors have added an explanation of the paper structure in the last paragraph of the introduction section.

7. [Methodology] Does figure 2 indicates the PDF?

As shown in Figure 3 and figure 4, the figures refer to the fuzzy membership function. In this study, triangular fuzzy membership function applied and explained in detail in Section 2.2. Figure 2 has been changed to Figure 3 in this revision.

8. [Methodology] In Figure 1, please describe the definition of                                               H?1, V?1, T?1.

The description has been added to Figure 2. Figure 1 has been changed to Figure 2 in this revision.

9. [Methodology] In Figure 4, please show full name for BDRY.

The “BDRY” has been replaced by “boundary” in Figure 5. Figure 4 has been changed to Figure 5 in this revision.

10. [Methodology] It is better to unify terms used in the study. For example, Breach and Failure in Figure 6 and Table 1.

The “Breach” has been replaced by “Failure” in Figure 7 and Table 1. Figure 6 has been changed to Figure 7 in this revision.

11. [Methodology] In Figure 7, please describe the definition of abbreviation (L, t and C).

The definitions of L, t and C have been added to the caption of Figure 8. Figure 7 has been changed to Figure 8 in this revision.

12. [Line 231] Are there any other parameters except the three parameters?

Although, there are several parameters such as failure height, failure side slope, failure start height, failure width, failure duration, and discharge coefficient,

This study considered three parameters among several parameters such as failure height, failure side slope, failure start height, failure width, failure duration, and discharge coefficient.

13. [Line 265] Table 1 should be changed to Table 2.

Typo has been corrected

14. [Table 2] What does the first column (e.g. Inundation analysis) of the table indicate? Four inundation analysis mean four scenarios? If so, why have you identified four scenarios, and what is the reason you assigned the two or three levees to each scenario?

The authors have partially revised Table 2 and he first column has been dropped in the Table 2

15. [Line 295] Fig. 2 should be changed to Figure 2.

Typo has been corrected

16. [Figure 5] Please re-plot this figure. I cannot see even right-side axis.

Figure 6 has been revised. Figure 5 has been changed to Figure 6 in this revision.

17. [Figure 10 and 11] Please add stream channels

The stream channel has been added to Figure 11 and 12. Figures 10 and 11 have been changed to Figure 11 and 12 in this revision.

18. [Figure 12] It is difficult to understand. Please add some background images or shape files which could help to understand the meaning of the results.

Figure13 shows the spatial representation of the triangular fuzzy membership function (a, b and c). So, we have modified the shape of the figure to be clear by adding a picture-specific outline rather than a background image. Figure 12 has been changed to Figure 13 in this revision.

19. [Figure 13] Please add flow direction.

The flow direction has been added to Figure 14. Figure 13 has been changed to Figure 14 in this revision.

20. [Figure 15] Please remove Korean.

We have removed all Korean from Figure 16, and added a representation of the 5th-grade areas derived from the flood hazard map created in this study. Figure 15 has been changed to Figure 6 in this revision.

21. [Conclusion] At this section, instead of repeating what authors did, describe the conclusions derived from the results.

We have adopted the reviewer’s suggestion and are pleased to present strong conclusions

The authors thank for the reviewer for carefully reviewing the paper and providing and assessment

Round 2

Reviewer 2 Report

Dear authors,

the revised version of the manuscript is really an improvement and thus can be recommmended to be published. I have only a few minor remarks:

Abstract Line 19: "[..] this study was compared [..]"

Line 59: I suggest to add the following reference to the list of references: Zischg, A. P., Felder, G., Weingartner, R., Quinn, N., Coxon, G., Neal, J., Freer, J., and Bates, P.: Effects of variability in probable maximum precipitation patterns on flood losses, Hydrol. Earth Syst. Sci., 22, 2759–2773, doi:10.5194/hess-22-2759-2018, 2018.

line 89: "[..] using a numerical model" or using numerical models

Figure 15: Please discuss in chapter 4.4 Review of Application Results why the floodplain east of the river and south of point (3) is not considered as flooded in your study while it does in the official hazard map

Author Response

Comments and Suggestions for Authors

Dear authors,

the revised version of the manuscript is really an improvement and thus can be recommmended to be published. I have only a few minor remarks:

The authors thank the reviewer for recommending a publication and understanding of the efforts we have made to improve the paper. Author Comments appear in BLUE, while Reviewer Comments appear in BLACK.

Abstract Line 19: "[..] this study was compared [..]"

The sentence has been revised.

Line 59: I suggest to add the following reference to the list of references: Zischg, A. P., Felder, G., Weingartner, R., Quinn, N., Coxon, G., Neal, J., Freer, J., and Bates, P.: Effects of variability in probable maximum precipitation patterns on flood losses, Hydrol. Earth Syst. Sci., 22, 2759–2773, doi:10.5194/hess-22-2759-2018, 2018.

We have expanded the citations as suggested by the reviewer. A reference has been added.

line 89: "[..] using a numerical model" or using numerical models

Typo noted and corrected

Figure 15: Please discuss in chapter 4.4 Review of Application Results why the floodplain east of the river and south of point (3) is not considered as flooded in your study while it does in the official hazard map

The reasons for the differences in the flood extents between the two maps of this study and MOLIT have been added to chapter “4.4 Review of Application Results”.